# QM/MM Modeling of the Flavin Functionalization in the RutA Monooxygenase

**DOI:** 10.3390/molecules28052405

**Published:** 2023-03-06

**Authors:** Bella Grigorenko, Tatiana Domratcheva, Alexander Nemukhin

**Affiliations:** 1Department of Chemistry, M.V. Lomonosov Moscow State University, Moscow 119991, Russia; 2N.M. Emanuel Institute of Biochemical Physics, Russian Academy of Sciences, Moscow 119334, Russia

**Keywords:** flavin, monooxygenases, RutA enzyme, molecular oxygen, quantum chemistry, QM/MM, absorption spectra

## Abstract

Oxygenase activity of the flavin-dependent enzyme RutA is commonly associated with the formation of flavin-oxygen adducts in the enzyme active site. We report the results of quantum mechanics/molecular mechanics (QM/MM) modeling of possible reaction pathways initiated by various triplet state complexes of the molecular oxygen with the reduced flavin mononucleotide (FMN) formed in the protein cavities. According to the calculation results, these triplet-state flavin-oxygen complexes can be located at both *re*-side and *si*-side of the isoalloxazine ring of flavin. In both cases, the dioxygen moiety is activated by electron transfer from FMN, stimulating the attack of the arising reactive oxygen species at the C4a, N5, C6, and C8 positions in the isoalloxazine ring after the switch to the singlet state potential energy surface. The reaction pathways lead to the C(4a)-peroxide, N(5)-oxide, or C(6)-hydroperoxide covalent adducts or directly to the oxidized flavin, depending on the initial position of the oxygen molecule in the protein cavities.

## 1. Introduction

Flavoprotein monooxygenases are involved in multiple metabolic processes [1]. The monooxygenase RutA is one of seven proteins on the Rut pathway, which are required by *Escherichia coli* (strain K12) to grow on uracil as the sole nitrogen source [2]. The flavoenzyme RutA directly cleaves the uracil ring between N3 and C4 to yield ureidoacrylate, as established by nuclear magnetic resonance spectroscopy and mass spectroscopy. However, a mechanism for this reaction is still a matter of discussion; in particular, it is required to explain how RutA incorporates an oxygen atom from O_2_ at the proper position of uracil. Flavoenzymes are able to control chemical reactions involving the oxygen molecule in the absence of metal or metal-containing cofactors. These enzymes exploit the properties of the flavin’s isoalloxazine ring to activate dioxygen and initiate productive reactions, often with the assistance of transient covalent flavin-oxygen adducts [3]. Recent results of experimental studies of RutA stimulated discussion of the factors that govern oxygenase activity in flavoenzymes, including the role of various flavin-oxygen adducts [4,5,6,7].

The active site in RutA stabilizes the reduced form of flavin mononucleotide (FMN), which is non-covalently bound in the solvent-accessible protein cavity. Figure 1 shows the reduced (Fl_red_) and oxidized (Fl_ox_) forms of FMN and the common numbering of atoms in the isoalloxazine ring. According to current knowledge [3,5,6,7], when the oxygen molecule is trapped in a protein cavity near the isoalloxazine ring, it is activated by an electron transfer from the reduced FMN. This leads to a pair of the superoxide radical (O_2_^−^) and the relatively stable semiquinone radical (Fl_SQ_), followed by the switch from the initial triplet spin state to the singlet spin state. The reactive C4a-N5 locus of the isoalloxazine moiety is usually considered a critical site for dioxygen (O_2_) activation and subsequent covalent adduct formation. These adducts, flavin-C(4a)-peroxide and flavin-N(5)-oxide, are recognized as the key oxygen-transferring intermediates in the catalytic mechanisms of the flavoprotein monooxygenases and oxidases [3].

According to the findings described in [6], the primary catalytic mechanism in RutA (as well as in the other flavoenzyme, EncM [8]) is associated with N5-oxigenation. Matthews et al. [6] conducted O_2_-pressurized X-ray crystallography at 5 or 15 bars and produced crystals of RutA that contained oxidized FMN and molecular oxygen (RutA–Fl_ox_–O_2_). The crystal structure PDB ID 6SGG was interpreted as the model containing the dioxygen moiety next to the N5 atom of the isoalloxazine ring with an unusually short distance of 2.09 Å between one of the oxygen atoms in O_2_ and the N5 atom of the flavin [6]. An important role of the amino acid residues Thr105 and Asn134 in forming the protein cavity at the *re*-side of the flavin cofactor was underlined upon dioxygen binding [6]. The crystal structure for the complex RutA–Fl_ox_–uracil was also produced (PDB ID 6SGL) [6]. The stopped-flow kinetics studies revealed that in the absence of uracil, the reduced form of FMN bound to RutA is converted to the oxidized form—the time-resolved absorption bands showed the gradual rise of the band with the maximum at 446 nm, typical for the protein-bound Fl_ox_ [8,9]. When uracil was added to the RutA–Fl_red_–O_2_ system, the spectra showed the mixed formation of Fl_ox_ and the flavin-N(5)-oxide adduct (Fl_N5O_); in particular, the band maximum was red-shifted to 463 nm, the value assigned to the Fl_N5O_ species [8].

The experimental work on RutA [6] and the related flavoprotein EncM [5] also included computational modeling of the reaction pathways associated with the formation of the covalent flavin-oxygen adducts. Ref. [5] contains the results of the density functional theory (DFT) calculations at the B3LYP-D3 level for the lumiflavin (reduced)—oxygen model system in the gas-phase. The computed profiles, leading to the Fl_C4aOO_^−^, Fl_N5OOH_, and Fl_N5O_ species, were corrected within the polarizable continuum model, taking into account the water solvent effects as well as the enthalpy and Gibbs energy corrections. The basic conclusion formulated by the results of the DFT modeling is that the formation of both flavin-C(4a)-peroxide (Fl_C4aOO_^−^) and flavin-N(5)-oxide (Fl_N5O_) is feasible. A year later, Luo and Liu [10] computed the energy profiles starting from the reduced lumiflavin and O_2_ and leading to the Fl_C4aOOH_ species in the gas phase (corrected within the continuum solvent models) using the DFT and complete active space self-consistent field (CASSCF) quantum chemistry methods.

To the best of our knowledge, no computational modeling of the formation of the products of the flavin-oxygen interaction inside the protein cavities in monooxygenases by the QM/MM method has been reported. Application of QM/MM calculations as compared with gas-phase-based simulations is important because the protein matrix provides a suitable cage to stimulate the triplet-singlet switch and recombination of the superoxide-Fl_SQ_ radical pair to a flavin-oxygen adduct before the radical intermediates diffuse apart [7]. Thus, the enzyme-mediated stabilization of the radical intermediates should be simulated in the protein environment.

In this work, we construct the model system RutA–FMN_red_–O_2_ using the coordinates of heavy atoms in the crystal structure PDB ID 6SGG [6]. The QM(DFT)/MM method is applied to locate several structures of complexes of the triplet state oxygen molecule with the molecular groups near the flavin’s isoalloxazine ring in the solvent-accessible protein cavities, i.e., the putative oxygen binding sites. The occurrence of different dioxygen positions in RutA was documented in our preliminary studies [11,12] based on molecular dynamics (MD) simulations. Here, we model the formation of products of the flavin-oxygen interaction initiated from various oxygen-binding sites.

Besides structures of the flavin-oxygen complexes in RutA, we also report here the computed excitation energies of the S_0_ → S_1_ and S_0_ → S_2_ transitions and the corresponding oscillator strengths (o.s.) of transient species, which provide estimates of the absorption band maxima in the electronic spectra. It is difficult to refer to large numbers of papers devoted to modeling the electronic spectra of flavin-based systems using a variety of quantum chemistry methods. We only note a recent review by Kar et al. [9], as well as the most relevant recent papers [13,14,15]. In this work, we apply the advanced excited-state method, the extended multi-configurational quasi-degenerate perturbation theory in the second order (XMCQDPT2) [16], which has proven its efficiency in predicting the excitation energies of organic chromophores [17,18,19,20].

## 2. Results and Discussion

In Figure 1, we show a general view of the model system is considered in this work. In the inset, we clarify the position of the isoalloxazine ring of FMN in the active site of the enzyme and the nearest molecular groups—the Asn134 and Thr105 side chains at the flavin’s *re*-side or water molecules at the *si*-side. We found several triplet state minimum energy structures of the complex of RutA with the oxygen molecule positioned near the isoalloxazine ring using the QM/MM method (see details in Section 3 below). Atomic coordinates of all located structures are available from the Appendix A for the present paper.

There are several important observations related to the computationally derived model systems containing the flavin-oxygen complexes. First, the optimized positions of dioxygen are located on both sides (*re*-side and *si*-side) of the isoalloxazine ring (see Figure 1). The sites at the *re*-side correspond to the oxygen pocket-1 noted in [12], as well as the position of oxygen in the X-ray structure PDB ID 6SGG [6]. The sites on the *si*-side, which correspond to the oxygen pocket-5 noted in [12], describe the oxygen molecule inside the shell of water molecules. Second, in each located oxygen-flavin complex, the dioxygen moiety should be identified as a superoxide O_2_^−^ anion rather than a neutral triplet oxygen molecule. Third, the energies of the triplet and singlet states in these flavin-oxygen complexes are almost degenerate. To support the second and third conclusions, we analyze the results of the complete active space self-consistent field (CASSCF) calculations of the complexes (for details, see Section 3 below). Thus, assuming a switch from the triplet state potential energy surface (PES) to the singlet state PES, modeling a triplet-singlet intersystem crossing, several pathways to the reaction products arise, including the formation of the flavin-oxygen adducts. We describe these pathways in the following subsections using the QM(DFT)/MM calculations and characterize the excitation energies of the products using the XMCQDPT2 calculations.

### 2.1. Pathway-1: Formation of the C(4a)-peroxyflavin Species from the Si-Side Oxygen Pocket

As described in the Introduction, the formation of the transient C(4a)-(hydro)peroxyflavin intermediates, Fl_C4aOOH_ or Fl_C4aOO_^−^, is commonly accepted in the chemistry of flavoenzymes [3,7], even though the C(4a)-peroxyflavin species in water is unstable and dissociates to hydrogen peroxide (H_2_O_2_) and the oxidized flavin [3]. Previous simulations [5,10] described the reaction pathways from reduced lumiflavin (Fl_red_) and O_2_ to Fl_C4aOO_^−^ and Fl_C4aOOH_ using the continuum solvent models. In both calculations, the transition from the initial triplet state model system to the singlet state system occurs after electron transfer in the configuration of the radical pair superoxide—Fl_SQ_. In both cases, low activation barriers of several kcal/mol were estimated.

According to the results of molecular dynamics simulations with the QM(DFT)/MM potentials for the RutA-O_2_-uracil model system [12], the C(4a)-peroxyflavin species was spontaneously formed after switching from the triplet state to the singlet state system at the trajectory frames that showed the occurrence of both dioxygen and uracil near the isoalloxazine ring at the *si*-side. Complexes of RutA with dioxygen located at the *si*-side pocket are located in this work using the QM/MM method (see Section 3 below for technical details). Figure 2a shows the triplet state minimum energy structure of the complex, which is denoted here as Complex-1. In this and other figures, we show critical distances (in Å) between heavy atoms and pay attention to the hydrogen-bond patterns. The water molecules, which are the components of hydrogen-bond networks, are numbered sequentially as they appear in the figures.

For Complex-1, we note that the Ox1-Ox2 distance in dioxygen increases from the value of about 1.20 Å in the gas-phase triplet state O_2_ molecule to 1.32 Å, thus favoring the formation of the superoxide anion. The latter is hydrogen bonded to the four nearest water molecules, with typical intermolecular distances between the oxygen atoms (2.6–2.9 Å). The C4a atom is the closest atom of the isoalloxazine ring to the dioxygen atoms, with a C4a-Ox1 distance 3.21 Å; the N5-Ox1 distance is 3.89 Å.

To characterize the electronic structure in Complex-1 in the triplet and singlet states, we carried out CASSCF(14/10)/cc-pVDZ calculations with the distribution of 14 electrons over 10 molecular orbitals. The latter include the orbitals located at the isoalloxazine ring and dioxygen (Figure 3). Orbitals 5–7 in both triplet and singlet states include notable contributions from dioxygen. Two dominant electronic configurations in the triplet Ψ_T_ and singlet Ψ_S_ wavefunctions are shown on the right side of Figure 3, indicating that the weight of the superoxide structure is considerable in the triplet state and dominant in the singlet state. Therefore, we conclude that Complex-1 bears the features of the Fl_SQ_–O_2_^−^ radical pair. Notably, the CASSCF energies of the triplet and singlet states (see the right side of Figure 3) are practically degenerate at this geometry since the energy gap is less than 0.1 kcal/mol. Therefore, we may assume that the switch from the triplet state PES to the singlet state PES in this model system occurs in the immediate vicinity of the geometry configuration of Complex-1.

Taking the structure of Complex-1 as a starting point for geometry optimization at the QM/MM level on the singlet state potential energy surface, we arrive at the structure shown in Figure 2b. In this system, the oxygen atom, Ox1, is covalently bound to C4a, leading to the distorted isoalloxazine ring. Another oxygen atom, Ox2, is hydrogen bonded to three water molecules; one water molecule from the initial structure, Wat2 in Figure 2a, drifted away from the dioxygen moiety. In agreement with the previous QM/MM studies of the reactions of the superoxide with the conjugated organic molecules inside proteins, e.g., with the chromophore of the green fluorescent protein [21,22], no potential energy barrier is detected on the pathway leading to the C(4a)-peroxyflavin species. The energy of the RutA structure with the C(4a)-peroxyflavin (Figure 2b) is 16.8 kcal/mol lower than the level of the reactants shown in Figure 2a.

It is instructive to compare these findings with the previously reported quantum chemistry computational results [5,10]. For the reaction of the reduced lumiflavin with the oxygen molecule, which starts at the triplet state PES and switches to the singlet state PES after an intersystem crossing (ISC) point, [10] reports the structure of the ISC point optimized at the CASSCF(12/9) level. The Ox1-Ox2 distance varies from the initial value of 1.20 Å for the reactants to the value of 1.31 Å near the ISC point to be compared to our 1.32 Å in Complex-1. The charge of the dioxygen moiety at this point is almost -1, as in the present work. The Ox1-Ox2 distance in the singlet state C(4a)-peroxyflavin adduct is 1.46 Å (versus 1.45 Å in the present work). The C4a-Ox1 distance varies from 3.14 Å (versus 3.21 Å in the present work) for the triplet state reactants to 1.41 Å (versus 1.36 Å in the present work) for the singlet state C(4a)-peroxyflavin adduct. The reaction energy profile shows the barrier of about 10 kcal/mol on the way from the initial triplet state structure of Fl^−^…O_2_ to the ISC point (the corresponding reaction segment is unavailable in our simulations). The reported potential energy difference between the ISC point and the singlet state C(4a)-peroxyflavin adduct (see Table S4 in [10]) for a water solution is 14.1 kcal/mol, to be compared with the value of 16.8 kcal/mol computed for the protein environment in the present work. We conclude that Complex-1 resembles the ISC point identified in [10]. In [5], the authors report in their Appendix A the computed potential energy difference of 8.1 kcal/mol between the C(4a)-peroxyflavin adduct and the reactants (Fl^−^…O_2_) for the water solution, using the B3LYP-D/6-311++G(2d,2p) calculations. The computed energy profile shows the energy barriers on both reaction steps: from (Fl^−^…O_2_) to Fl_SQ_…O_2_^−^ and then to the Fl_C4aOO_^−^ adduct.

With respect to the S_0_ → S_1_ excitation, our XMCQDPT2 calculations (see details in Section 3 below) report the excitation energy of the model system (Figure 2b) at 3.47 eV (357 nm, 0.57 o.s.), in excellent agreement with the experimental absorption band known for the C(4a)-peroxyflavin intermediate [3].

To conclude this subsection, we emphasize that the formation of the C(4a)-peroxyflavin intermediate is a commonly accepted route in monooxygenase activity, as reproduced in our calculations. We report an interesting finding that the oxygen molecule approaches the C4a position from the *si*-side pocket, i.e., from the site, presumably occupied by a uracil substrate in RutA [6]. Simulations described in [12] show that location of both the substrate and dioxygen at the same site of the isoalloxazine ring does not prevent the formation of Fl_C4aOO_^−^; this intermediate may occur and attack the substrate. Few theoretical papers report calculations of the functionalized flavins at the position C4a, e.g., refs. [23,24].

### 2.2. Pathway-2: Oxygen-N5 Interaction from the Si-Side Oxygen Pocket

Another triplet-state complex of the reduced flavin with dioxygen located at the *si*-side pocket is shown in Figure 4a. The QM/MM energy of this Complex-2 is 1.5 kcal/mol lower than the energy of Complex-1, indicating that these structures may be almost equally populated. Complex-2 shares the same features with Complex-1 in the sense that the superoxide describes the dioxygen moiety in this environment—the Ox1-Ox2 distance is 1.30 Å, and the charge on dioxygen is almost -1. However, in Complex-2, dioxygen stays closer to the isoalloxazine ring—the Ox2-N5 distance (2.58 Å) is about 0.6 Å shorter than the Ox1-C4a distance of 3.21 Å in Complex-1. Complex-2 also shows different hydrogen-bonding patterns as compared to Complex-1 (cf. Figure 2a and Figure 4a)—only two water molecules, Wat4 and Wat5 in Figure 4a, are within the hydrogen-bond distances from the dioxygen atoms; the water molecule Wat6 stays further from the active site, located on the way to bulk solvent.

Analysis of the electronic structure in Complex-2 in the triplet and singlet states using CASSCF(14/10) calculations was carried out similarly to that described in the preceding subsection for Complex-1. Like the previous case, the dominant electronic configuration corresponds to the superoxide structure, and the energy gap between the triplet and singlet states in Complex-2 is small (about 1.6 kcal/mol).

The system shown in Figure 4b corresponds to the minimum energy structure at the singlet state PES obtained in unconstrained QM/MM minimization initiated from the structure of Complex-2. The hydroperoxide anion HO_2_^−^ (hydrogen-bonded to water molecules Wat4, Wat5) is obtained as a result of a barrier-less transfer of the hydrogen atom transfer from N5. The QM/MM energy of the model system shown in Figure 4b is 7.8 lower than the energy of Complex-2.

Subsequent molecular events may develop along different scenarios. First, if a substrate is present in the protein cavity, the hydroperoxide anion may serve as an oxidative agent. Second, a proton from the neighboring water molecules (here, Wat5 or Wat4) may be transferred to the HO_2_^−^ species, leading to the hydrogen peroxide molecule (see Figure 4c for the scenario with Wat5 as a proton donor). The obtained hydroxyl anion (here, in the vicinity of the isoalloxazine ring) may serve as the oxidative agent if a substrate is located nearby. Third, a more extended chain of proton transfer events may take place over proton wires, borrowing a proton from bulk, which finally removes the charged species away from the oxidized FMN. Such proton wires involving a number of water molecules are present in this system (see Figure 1). Proton transfer events over the aligned chains of water molecules are characterized by low energy barriers within 10 kcal/mol [25]; thus, these pathways are feasible.

The S_0_ → S_1_ and S_0_ → S_2_ excitation energies computed at the XMCQDPT2 level at the configuration shown in Figure 4b are 2.46 eV (503 nm, 0.20 o.s.) and 3.60 eV (344 nm, 0.13 o.s.). The absorption band maxima are close to those expected for the oxidized flavin in the protein environment. We conclude that pathway-2 describes the mechanism of flavin oxidation Fl_red_^−^ + O_2_ (+H_2_O) → Fl_ox_ + H_2_O_2_ +OH^−^ in RutA in the absence of the substrate via the occurrence of the transient hydroperoxyl species.

### 2.3. Pathway-3: Oxygen-N5 Interaction from the Re-Side Oxygen Pocket

The scenario described in the preceding subsection (pathway-2) is also an option when starting from one more triplet-state flavin-oxygen complex, but this time, the oxygen molecule is located at the *re*-side pocket. The occurrence of the oxygen molecule at the *re*-side is detected in the X-ray structure PDB ID 6SGG [6]. According to the results of the crystallography analysis, the distance between the N5 atom of flavin and the nearest oxygen atom of O_2_ is unusually short—2.1 Å, whereas the other oxygen atom is coordinated by the nitrogen atom of the Asn134 side chain (the distance is 2.6 Å) and the oxygen atom of the Thr105 side chain (the distance is 2.7 Å) [6]. The oxygen-binding pocket at the *re*-side was also characterized in our previous simulations [11,12], although no such short nitrogen-oxygen distance as 2.1 Å was observed in these simulations. Another important result of [12] is that the switch from the triplet state PES to the singlet state PES at some MD trajectory frame of the flavin-oxygen complex led to the barrier-less formation of the hydroperoxide anion HO_2_^−^, similar to what we describe in the preceding subsection of this paper.

The structure of the triplet state flavin-oxygen complex with O_2_ from the *re*-side as obtained in QM/MM optimization in the present work (Complex-3) is shown in Figure 5a. We note that the dioxygen moiety again should be assigned to a superoxide O_2_^−^; the Ox1-Ox2 distance is 1.31 Å and the charge of the dioxygen is close to -1. The geometry configuration differs from the structure PDB ID 6SGG [6]—the dioxygen is hydrogen-bonded to flavin, Thr105, and water, whereas the Ox1-N(Asn134) distance is considerably longer (3.3 Å vs. 2.6 Å in the crystal), as well as the Ox1-N5 distance (2.6 Å vs. 2.1 Å in the crystal).

The results of the CASSCF(14/10) analysis of the electronic structure of Complex-3 (see Figure 6) show that the dominant electronic configuration in the triplet state multiconfigurational function corresponds to the superoxide structure of dioxygen, whereas the singlet state configuration features an even higher electron population in the dioxygen orbitals. The energy gap between the triplet and singlet states in Complex-3 computed at the CASSCF level is small (1.1 kcal/mol), similar to all flavin-oxygen complexes in the active site of RutA.

At least two minimum energy structures can be located on the singlet state PES using the QM/MM calculations. The structure shown in Figure 5b is almost isoenergetic with the triplet state Complex-3. Consistent with pathway-2 (see Figure 4b), the dioxygen borrows a hydrogen atom from N5 to Ox1. In pathway-3, the increased electron population on dioxygen leads to the spontaneous formation of hydrogen peroxide at the expense of a nearby water molecule, Wat7. The structure shown in Figure 5c is 6.8 kcal/mol lower in energy than the initial Complex-3. The chain of water molecules provides a suitable proton wire to move the negative charge of the hydroxyl farther away from the isoalloxazine ring.

The S_0_ → S_1_ and S_0_ → S_2_ excitation energies computed at the XMCQDPT2 level for the system shown in Figure 5b are 2.46 eV (503 nm, 0.20 o.s.) and 3.64 eV (344 nm, 0.20 o.s.). They are almost the same as the reaction products at pathway-2 (see Figure 4b) and close to those of the oxidized flavin in the protein environment. We conclude that pathway-3 (as well as pathway-2) describes the mechanism of flavin oxidation Fl_red_^−^ + O_2_ (+H_2_O) → Fl_ox_ + H_2_O_2_ + OH^−^ in RutA in the absence of a substrate.

### 2.4. Pathway-4: Formation of the N(5)-oxide Adducts

An intriguing pathway-4 also starts from the triplet state flavin-oxygen complex (Complex-4) at the *re*-side, although it should be noted that its QM/MM energy is 5.6 kcal/mol higher than the energy of Complex-3. The structure shown in Figure 7a differs from the previously described Complex-3 by the arrangement of the dioxygen—the Ox1-N5 distances are the same (2.62 Å) in both cases, but the Ox2 atom points towards Wat7 in Complex-3 and towards Asn134 in Complex-4. The latter structure shows poor coordination of dioxygen by the Thr105 side chain—the distance of 3.80 Å between Ox1 and the hydroxyl oxygen of Thr105 is long enough. Like all previously described flavin-oxygen complexes, Complex-4 accommodates the superoxide anion in the protein cavity—the Ox1-Ox2-distance is 1.32 Å; the charge of dioxygen is close to -1; and the energy gap between the triplet and singlet states is small.

Switching to the singlet-state PES at this point and subsequent unconstrained geometry optimization led to the structure (Figure 7b) with the hydroperoxyl and Ox1-Ox2 distance of 1.44 Å. The energy of this structure is 12.8 kcal/mol lower than the level of the initial Complex-4. It is important to note the following distances: Ox2-N5 (2.88 Å) and Ox1-O(Wat7) (2.72 Å). We note in passing that all QM/MM optimized structures in both triplet and singlet states of model systems do not reveal a short oxygen-nitrogen distance of 2.09 Å between any of the dioxygen atoms and the N5 flavin atom reported in the crystal structure PDB ID 6SGG [6]; no values less than 2.5 Å were obtained in simulations.

A gradual decrease of the Ox2-N5 distance results in the formation of the N-O covalent bond with a simultaneous break of the Ox1-Ox2 bond and finally leads to the formation of the flavin-N(5)-oxide, as shown in Figure 8a. This N5-O adduct is responsible for flavin functionalization in the flavoenzyme EncM [5,6]. According to the present QM/MM calculations, its energy is 28.5 kcal/mol lower than the energy of Complex-4. This agrees with the results of quantum chemistry gas-phase calculations of the reaction of the reduced lumiflavin with the oxygen molecule leading to the lumiflavin-N(5)-oxide described in [5]—the computed reaction energy is about −23 kcal/mol. Moreover, the reaction scheme depicted in Figure 4 in [5] seems consistent with the mechanism revealed in the present QM/MM simulations of the protein. As shown in Figure 7b, the hydroxyl Ox1-H^−^ formed after the cleavage of the Ox1-Ox2 bond interacts with the water molecule Wat7, leading to a newly formed water molecule Wat9 and a hydroxyl anion.

The S_0_ → S_1_ and S_0_ → S_2_ excitation energies computed at the XMCQDPT2 level at the configuration shown in Figure 8a are 2.44 eV (508 nm, 0.12 o.s.) and 3.25 eV (381 nm, 0.16 o.s.). These values may be compared to the absorption band maxima of 463 nm (2.69 eV) and 360–380 nm (3.26–3.44 eV) of the flavin-N(5)-oxide reported for the EncM protein (see Figure S5 in [5]). Taking into account possible errors in calculations of excitation energies in complex systems [26], the agreement is reasonable; moreover, the presence of the charged species OH^−^ near the chromophore in our model system (Figure 8a) may cause some shifts in the computed excitation energies.

As shown in Figure 8a,b, the N(5)-oxide undergoes a further transformation, namely, the hydroxyl emerged from Wat7 (Figure 7b) attaches a proton from Wat8, stimulating the newly derived hydroxyl (from Wat8) to attack flavin at the C8 position to form a new covalent bond. This adduct refers to the C(8)-hydrated N(5)-oxide compound shown in Figure 8b. These two structures are almost degenerate in energy. The hydration reaction of a chromophore in photoactive proteins upon light illumination is known, e.g., in the Dreiklang protein [27,28,29]; here, this reaction may occur at the ground singlet state PES.

The S_0_ → S_1_ and S_0_ → S_2_ excitation energies computed at the XMCQDPT2 level for the system shown in Figure 8b are 2.73 eV (454 nm, 0.38 o.s.) and 3.54 eV (350 nm, 0.12 o.s.). We see a considerable change in the S_0_ → S_1_ band (from 508 nm to 454 nm) due to chromophore hydration. This shift is clear as the electronic structure of flavin is closer to the Fl_ox_ for the flavin-N(5)-oxide and closer to the Fl_red_ for the flavin- C(8)-hydrated N(5)-oxide. The chemical formulas of both species are given in Figure 2. We note a change in the conjugation patterns consistent with the equilibrium geometry structures (see, e.g., the distances between N5 and carbon atoms in Figure 8).

### 2.5. Pathway-5: Formation of the C(6)-hydroperoxyflavin Species from the Si-Side Oxygen Pocket

Finally, we describe an uncommon pathway in flavin functionalization that explores the C6 position of the isoalloxazine ring. Spontaneous covalent binding of the hydroperoxide at the C6 atom was noted in the molecular dynamics simulations with the QM/MM potentials for the RutA-O2-uracil model system [12]. The occurrence of the reaction intermediates with the C6-O bond formation was discussed in the recent study [30].

Figure 9a shows a fragment of the triplet-state model system (Complex-5) formed from the *si*-side oxygen pocket. The energy of this structure is 5.4 kcal/mol higher than that of Complex-1, which initiates the formation of the flavin-C(4a) adduct, and is 6.9 kcal/mol higher as compared to Complex-2, which gives rise to the oxidized flavin and hydrogen peroxide.

Complex-5 shows the features common to all flavin-oxygen complexes considered in the present work: the Ox1-Ox2 distance is 1.31 Å; the charge on dioxygen is close to -1; the triplet-singlet gap is small; and the dioxygen moiety is hydrogen bonded to the nearest molecular groups, in this case, to four water molecules: Wat6, Wat7, Wat8, and Wat11. The shortest distance to the atoms of the isoalloxazine ring refers to the Ox1-C6 separation—3.02 Å.

Starting from Complex-5, a structure shown in Figure 9b with the flavin-C(6)-hydroperoxide adduct (noted in the previous QM/MM MD simulations [12]) is readily obtained after switching from the triplet state PES to the singlet state PES as a result of the forming covalent bond C6-Ox1 coupled with proton transfer from N5 to Ox2. The energy of this structure is 6.5 kcal/mol lower than that of Complex-5. Principally, this pathway may proceed further; a potential barrier of about 14 kcal/mol is required to break the Ox1-Ox2 bond and to arrive at the structure shown in Figure 9c with the C(6)-C(7)-epoxide. Such cyclic patterns were noted earlier in the modeling reactions of oxygen with the chromophore of the green fluorescent protein [22]. The energy of the thus obtained structure (Figure 9c) is 30.1 kcal/mol lower than that of Complex-5, and this adduct may occur as a possible reaction intermediate. Its S_0_ → S_1_ excitation energies computed at the XMCQDPT2 level is 2.57 eV (483 nm, 0.06 o.s.).

### 2.6. Summary and Concluding Remarks

In this work, we apply QM/MM and quantum chemistry methods to characterize minimum energy points on the triplet state and singlet state potential energy surfaces along the possible reaction pathways initiated by the interaction of the oxygen molecule with the molecular groups in the protein cavities of the flavoenzyme RutA. Table 1 summarizes the most important results of these simulations, grouped as the oxygen activation pathways, starting from the oxygen-containing pockets at the *si*-side and *re*-side of the flavin’s isoalloxazine ring.

In Table 1, we compare the energies of the initial triplet state flavin-oxygen complexes (column 2) and of the corresponding reaction products (column 4) differently for the structures emerging from either the *si*-side or *re*-side oxygen pockets, despite the fact that these model systems were created for the same QM-MM partitioning scheme described below in Section 3. As clarified in Figure 1, the *re*-side region is deeper in the protein macromolecule, whereas the *si*-side is entirely in the solvent-accessible cavity. It is practically impossible to provide fully equivalent conditions for the QM/MM geometry optimization in these two regions because of the different impacts of the large amount of water molecules in the MM subsystem. For instance, the total QM/MM energies for the *re*-side initial flavin-oxygen systems (Complexes-3 and 4) are about 15 kcal/mol lower than the energies of the *re*-side initial complexes.

Analysis of the data collected in Table 1 allows us to conclude that pathways 1, 2, and 3 may refer to the most probable reaction routes leading to oxygen activation in RutA if the estimated energy barriers are considered. Pathways 2 and 3 are practically similar despite the different starting positions of the oxygen molecule—from the protein interior or from the solvent-accessible cavity. These routes describe the formation of the oxidized flavin, hydrogen peroxide, and the hydroxyl anion with a reaction energy of 7–8 kcal/mol. The computed absorption band maxima (the last column in Table 1) are the same in the two pathways showing characteristic features of the oxidized flavin. Pathway-1 leading to the formation of the flavin-C(4a)-peroxide well-known from experimental studies is also highly likely, as validated by the computed relative energies and absorption spectra.

Pathways 4 and 5 are initiated from the structures, which may be less populated considering the computed energies (6–7 kcal/mol above the level of the lowest-energy structures on the respective side) of Complex-4 and Complex-5. However, the products of the corresponding reactions, whose energies are low enough relative to the level reactants, are very interesting and describe the known [3,4,5,6,7] and newly characterized covalent flavin-oxygen adducts.

The computed energies of the reaction products (column 4 in Table 1) are large enough to expect that the corresponding species may be detected experimentally. The computed excitation energies and oscillator strengths are reported for every reaction product, providing useful data to analyze transient absorption bands in prospective spectral studies.

Next, we emphasize the essential role of water molecules and of proton transfer along hydrogen bond networks involving water molecules in the solvent-accessible protein cavities of RutA. This refers to all pathways studied here, and it is important to note that proton transfer over the aligned chains of water molecules is characterized by low energy barriers—within 10 kcal/mol, e.g., according to [25].

Finally, we comment that the energy profiles connecting these minimum energy points are roughly estimated in the present QM/MM calculations. We point out that there are unavoidable errors in the estimates of the activation barriers on reaction routes calculated by the QM/MM method with the modern quantum chemistry software packages, amounting to several kcal/mol [31].

## 3. Models and Methods

Model molecular systems were created as follows: The coordinates of heavy atoms were taken from the crystal structure PDB ID 6SGG [6] containing the oxidized FMN. We added hydrogen atoms using molecular mechanics tools, assuming that the side chains of Arg and Lys were positively charged and that the side chains of Glu and Asp were negatively charged. The reduced form of FMN was constructed. The oxygen molecules were added to the protein cavities near the isoalloxazine ring. The visual molecular dynamics (VMD) program [32] was applied to build the solvation water shells shown in Figure 1. Classical molecular dynamics simulations were carried out with the NAMD program [33] using the CHARMM36 force field [34] to equilibrate the system composed of more than 42,000 atoms. The charge of the reduced FMN species was -3, whereas the charge of the entire model system was -4.

In QM/MM optimization, a large fraction of the active site was assigned to the QM-part (Figure 10). Specifically, all atoms of FMN with the phosphate group, the oxygen molecule, the side chains of Thr105, Asn134, Trp139, Glu292, and 19 water molecules were included in the QM subsystem, which comprises 150 atoms. The Thr105 and Asn134 side chains contribute to the coordination of the dioxygen at the *re*-side; the Trp139 and Glu292 side chains border the solvent-accessible protein cavity at the *si*-side. Four hydrogen link atoms were added to the carbon atoms highlighted in yellow in Figure 10 to saturate the broken covalent bonds C_α_-C_β_ for Thr105, Asn134, Trp139, or the C_β_-C_γ_ bond for Glu292.

Optimization of the geometry parameters and calculation of energies were performed with the NWChem software package [35]. The density functional theory PBE0 functional [36] with the D3 correction [37] and the 6–31G* basis set were used in the QM part, whereas the AMBER99 force field parameters [38] were applied to describe the MM subsystem. The electrostatic embedding scheme was used to polarize the QM region by the Coulomb potential due to MM charges (see the relevant discussion in [31]). The unrestricted DFT approach was used in QM/MM calculations of energies and forces for the triplet state model systems, and the restricted DFT approximation was applied for the singlet state systems. No constraints were imposed in the geometry optimization of minimum energy stationary points. Crude estimates of the energy profiles near possible barriers were carried out using constrained minimization.

The state-averaged CASSCF method with the distribution of 14 electrons over 10 complete active orbitals represented by the cc-pVDZ basis set was used to analyze the electronic structure of the flavin-oxygen complexes for a large molecular cluster identical to the QM subsystem in QM/MM simulations. To select the CASSCF scheme, we carried out a series of preliminary calculations at the configuration interaction level to recognize the orbitals that must be assigned to the active space. Further enlargement of the active space is not technically feasible.

The optimized active orbitals for Complex-1 and Complex-3 are illustrated in Figure 3 and Figure 6. Calculations of the vertical excitation energies and oscillator strengths were carried out using the extended multiconfigurational quasi-degenerate perturbation theory in the second order (XMCQDPT2) [16] based on the SA-CASSCF(14/10) orbitals. All calculations for cluster models were performed using the Firefly package [39].

## 4. Conclusions

We show that various location sites of the oxygen molecule in the solvent-accessible protein cavities initiate various reaction pathways of flavin functionalization in the flavoenzyme RutA in the absence of the substrate. Several triplet state complexes of the molecular oxygen with the reduced flavin mononucleotide are optimized at both the *re*-side and *si*-side of the isoalloxazine ring of flavin using the QM/MM method. In every complex, the dioxygen moiety in the protein cavity is best described as the superoxide anion O_2_^−^ interacting with the flavin semiquinone. Reactions on the singlet state potential energy surface can follow several pathways. Along these pathways, the formation of the following products occurs: oxidized flavin and hydrogen peroxide, or covalent flavin-oxygen adducts such as C(4a)-peroxide, N(5)-oxide, or C(6)-hydroperoxide. Novel variants of the adducts—C(8)-hydrated N(5)-oxide and C(6)-hydroperoxide—are characterized as the minimum energy structures in the model protein-oxygen system. Excitation energies and oscillator strengths are estimated for the reaction products. The results of the present simulations contribute to the studies of molecular processes of flavin functionalization and the factors that govern oxygen reactivity in flavoenzymes, taking RutA as an important example.

## Data Availability

The files with atomic coordinates of the stationary points on potential energy surfaces in the pdb-format are deposited to the general-purpose open-access repository ZENODO, which can be accessed via https://doi.org/10.5281/zenodo.7558382 (accessed on 26 January 2023).

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
