# Peer review of "QM/MM Modeling of the Flavin Functionalization in the RutA Monooxygenase"

_molecules, 2023, doi:10.3390/molecules28052405_

Round 1

Reviewer 1 Report

Grigorenko et al. employ QM/MM calculations to study the binding of molecule oxygen in RutA-FMN. This is a comprehensive study of the different ways oxygen can potentially react with the flavin in RutA, depending on its initial placement. This study attests to the remarkable versatility of flavin and its ability to undergo different chemical transformations under slightly different conditions.

I found this work technically sound and very interesting. There is an aspect of mechanisms 2 and 3 that are important to clarify before the manuscript is accepted for publication. Once those are addressed, the manuscript would be acceptable for publication in Molecules in my view.

Specifically, starting with mechanism 2.2. The product is indicated as the fully oxidized flavin. This means there must be another electron transfer step involved from the semiquinone intermediate to the fully oxidized product. The description on page 7 is unclear about this. If flavin is indeed fully oxidized, then HO2 is a negatively charged hydroperoxide anion (rather than hydroperoxyl radical), and the preceding step was a hydrogen atom transfer or PCET (rather than a proton transfer). Without some electron transfer, flavin would still be a semiquinone. Note that the anionic semiquinone has somewhat similar absorption wavelengths as the oxidized flavin, so it is important to check where the charge resides to confirm the identity. I suppose this can be confirmed by checking the charge on the FMN and the HO2 in the Fib. 4b structure.

The same is true for mechanism 2.3 (page 9); if FMN is indeed fully oxidized, then a hydrogen atom transfer or PCET is involved, rather than a proton transfer.

Author Response

We thank the Reviewer for pointing out the mistake. The HO2 species is a negatively charged hydroperoxide anion. We introduced corrections at five places (marked red in the highlighted version) of the revised manuscript.

Reviewer 2 Report

In the present manuscript entitled "QM/MM Modeling of the Flavin Functionalization in the RutA Monooxygenase", by B. Grigorenko et al. five different pathways for the functionalization of flavin are scrutinized. The results of the present work could be used by experimentalists as a guide to finding out the mechanism and explaining crystallized structures. Although the different sections of this work are understandable, there are several parts that could be improved before I could consider it for publication.  Here there is a list of these parts divided into major and minor issues:    Major issues   1. The introduction could be improved by explicitly writing what the function of the enzyme considered in this study is. Why is it interesting to learn more about this enzyme in general terms?   It seems to me that the structures considered in the present work refer to the already excited ones, but in nature, how are these states generated?   2. In the Results and Discussion section:   * how the initial complexes were obtained? what was the reasoning behind choosing those complexes? * on lines 173-174 an energy gap of 0.1 kcal/mol is mentioned, is this mentioned, however, my calculation gives the following value:   (2537.564711-2537.564054)*627.52 =0.412 kcal/mol what value is the correct one?   * On the figures showing the orbitals such as Fig. 3, could you write the weights for each of the 10 configurations maybe inside the boxes?    * On line 189: "no potential energy barrier is detected" did the authors perform a sampling along some reaction coordinate and obtained the PES? Otherwise, how could they know that there is no energy barrier?    3. In the Summary and concluding remarks section:   * on line 442 it reads "... allows us to conclude that pathways 1,2,3 may refer to the most probable reaction routes ..." what is the reasoning behind this claim? From Table 1, it seems to me that pathways 4 and 5 had lower energies at the product side so they could be more stable than the others?    * on line 468: "There is a little sense to carry out expensive calculations to locate transition states ..." As I am reading this sentence, it seems to me that you meant that because the modelling of QM/MM in gas phase is an accepted protocol, transition states are not important? A better explanation is needed at this point.    4. In the Models and Methods section:   * Please explain clearly all the types of simulations you did. Separate Classical MD (if used), QM/MM, and pure QM simulations with details for each section.    * Explain where each type of simulation was used   * Was the system equilibrated?    * How did you generate the initial complexes?   * For QM/MM and QM, what was the total charge of the QM region? What scheme did you use for breaking covalent bonds for residues in the QM region? What was the total charge of the system?    * For purposes of reproducibility, could you include the simulation scripts for the systems and software considered in this work?    * Could you provide more details of why for CASSCF 14 electrons over 10 active orbitals scheme was chosen?    * The values for excitation energies S0->S1 are reported but very little is said about the physical meaning. Could you extend these descriptions?      Minor issues:   * Grammar needs to be checked, for instance on line 85: "... by motifs of the crystal ..." maybe this can be rephrased

Author Response

Please see tha attachment

Round 2

Reviewer 2 Report

The authors went through my comments and addressed them adequately. The Introduction is more concise, and the methods and results are now clearly explained. For these reasons, I accept the manuscript as it is in its present form.